# Matrix Vesicle-Mediated Mineralization and Osteocytic Regulation of Bone Mineralization

**DOI:** 10.3390/ijms23179941

**Published:** 2022-09-01

**Authors:** Tomoka Hasegawa, Hiromi Hongo, Tomomaya Yamamoto, Miki Abe, Hirona Yoshino, Mai Haraguchi-Kitakamae, Hotaka Ishizu, Tomohiro Shimizu, Norimasa Iwasaki, Norio Amizuka

**Affiliations:** 1Developmental Biology of Hard Tissue, Graduate School of Dental Medicine, Faculty of Dental Medicine, Hokkaido University, Sapporo 060-8586, Japan; 2Northern Army Medical Unit, Camp Makomanai, Japan Ground Self-Defense Forces, Sapporo 005-8543, Japan; 3Division of Craniofacial Development and Tissue Biology, Graduate School of Dentistry, Tohoku University, Sendai 980-8577, Japan; 4Orthopedic Surgery, Faculty of Medicine, Hokkaido University, Sapporo 060-8638, Japan

**Keywords:** bone mineralization, osteoblast, osteocyte, matrix vesicle, Phex/SIBLING

## Abstract

Bone mineralization entails two mineralization phases: primary and secondary mineralization. Primary mineralization is achieved when matrix vesicles are secreted by osteoblasts, and thereafter, bone mineral density gradually increases during secondary mineralization. Nearby extracellular phosphate ions (PO_4_^3−^) flow into the vesicles via membrane transporters and enzymes located on the vesicles’ membranes, while calcium ions (Ca^2+^), abundant in the tissue fluid, are also transported into the vesicles. The accumulation of Ca^2+^ and PO_4_^3−^ in the matrix vesicles induces crystal nucleation and growth. The calcium phosphate crystals grow radially within the vesicle, penetrate the vesicle’s membrane, and continue to grow outside the vesicle, ultimately forming mineralized nodules. The mineralized nodules then attach to collagen fibrils, mineralizing them from the contact sites (i.e., collagen mineralization). Afterward, the bone mineral density gradually increases during the secondary mineralization process. The mechanisms of this phenomenon remain unclear, but osteocytes may play a key role; it is assumed that osteocytes enable the transport of Ca^2+^ and PO_4_^3−^ through the canaliculi of the osteocyte network, as well as regulate the mineralization of the surrounding bone matrix via the Phex/SIBLINGs axis. Thus, bone mineralization is biologically regulated by osteoblasts and osteocytes.

## 1. Introduction

Bone is mineralized tissue composed of crystalline calcium phosphates and collagen fibrils onto which the calcium phosphate crystals are deposited [1,2,3,4]. The hardness and flexibility of bone, which can provide strength against mechanical force, are derived from calcium phosphates and collagen fibrils, respectively. Osteoblasts secrete a large amount of collagen fibrils, non-collagenous proteins, and proteoglycans, as well as matrix vesicles, into the incompletely mineralized superficial layer of the bone matrix known as the osteoid. A matrix vesicle is a small extracellular vesicle equipped with membrane transporters and enzymes involved in mineralization, such as tissue nonspecific alkaline phosphatase (TNAP) [5,6], ectonucleotide pyrophosphatase/phosphodiesterase (ENPP) [7], sodium-dependent phosphate cotransporter type III (Slc20a1/Pit1 and Slc20a2/Pit2) [8,9,10,11,12], phosphoethanolamine phosphohydrolase 1 (PHOSPHO1) [13,14,15], and ankylosis (ANK) [16,17]. The supply and inflow of calcium ions (Ca^2+^) and inorganic phosphate ions (PO_4_^3−^) in the initial process of matrix vesicle-mediated mineralization is achieved by the finely-tuned activities of these enzymes and transporters. Hence, matrix vesicle-mediated mineralization is categorized as the primary mineralization that takes place in the osteoid, which then forms a mineralized nodule, also called a calcifying nodule, allowing the collagen mineralization to eventually spread throughout the bone [18,19]. Moreover, non-collagenous proteins and proteoglycans in the osteoid regulate mineralization by modulating the aggregation of collagen fibrils and mineral deposition during primary mineralization [20,21,22,23].

After primary mineralization, the bone mineral density becomes slowly and chronologically elevated in a phenomenon called secondary mineralization, which is independent of osteoblastic bone formation [2]. It is hypothesized that secondary mineralization is achieved by the physicochemical processes of mineral transport in the osteocytic network extended throughout the bone [24]. Therefore, the osteocytes and the meshwork of their cytoplasmic processes appear essential for secondary mineralization involving bone mineral transportation [25,26,27,28]. Osteocytes with well-organized osteocytic lacunar canalicular system (OLCS) sense mechanical stress [29], transport bone minerals, and secrete bone metabolism-regulating molecules [30,31,32,33,34]. It is important that the cytoplasmic processes of osteocytes connect with those of the bone-forming osteoblasts and the narrow channels through which osteocytic processes are opened to the osteoid where the matrix vesicle-mediated mineralization takes place. Therefore, it is postulated that bone minerals, such as Ca^2+^ and PO_4_^3−^, are derived from the activities of TNAP/ectonucleotide pyrophosphatase/phosphodiesterase 1 (ENPP1) and PHOSPHO1 inside the matrix vesicles and also from the osteocytic canaliculi that are opened to the osteoid, implicating the interplay of osteoblasts and osteocytes for adequate maintenance of mineralized bone.

In this review, we will discuss matrix vesicle-mediated mineralization, osteocytic regulation of bone mineralization, and the possible interplay of osteoblasts and osteocytes on bone mineralization.

## 2. Matrix Vesicle-Meditated Mineralization

### 2.1. Nucleation of Calcium Phosphates in Matrix Vesicles

Bone mineralization initiates inside matrix vesicles, which are small extracellular vesicles secreted by osteoblasts [18,19,35,36,37]. Matrix vesicles contain several membrane transporters and enzymes related to mineralization on their plasma membranes and in their interior, thus providing an adequate microenvironment for calcium phosphate nucleation and subsequent growth to eventually form hydroxyapatite crystals [Ca_10_(PO_4_)_6_(OH)_2_]. Regarding calcium phosphate crystal nucleation, it is generally known that Ca^2+^ strongly binds to the negatively charged inner leaflet of the plasma membrane [38]. Plasma membranes consisting of phosphatidylcholine and phosphatidylserine have a substantial capacity for Ca^2+^ binding, with several types of binding sites proposed [39]. Many researchers have previously postulated that phosphatidylserine, which is rich in matrix vesicles, has a particularly high affinity for Ca^2+^ to produce a stable calcium phosphate-phospholipid complex with the vesicle’s membrane [40,41,42]. When observed by transmission electron microscope (TEM) during the early phases of calcium phosphate nucleation, amorphous non-crystalline calcium phosphates are found to incipiently form in association with the inner leaflet of the matrix vesicle membranes [43]. Thereafter, the original amorphous calcium phosphate structures are converted into mature crystalline structures, i.e., hydroxyapatite, allowing the large, assembled mineral crystals inside the matrix vesicles. The resultant hydroxyapatite crystals grow in all directions inside the vesicle, then penetrate the plasma membrane, and eventually form mineralized nodules in the later stage (Figure 1).

### 2.2. Distribution of Ca and P in the Vicinity of Matrix Vesicles in the Osteoid 

Electron energy loss spectroscopy, or EELS, enables elemental mapping and can detect calcium (Ca) and phosphorus (P) at an ultra-structural level [45]. A previous report revealed that Ca and P were highly accumulated inside matrix vesicles; however, Ca was evenly and abundantly distributed in the vicinity of the matrix vesicles, while P was detected predominantly in organic materials, such as collagen fibrils and cells, but not in the matrix vesicle vicinity [45]. Nucleation and growth of calcium phosphate crystals require the influx of Ca^2+^ and PO_4_^3−^ inside the matrix vesicles from the extracellular fluid. Taking Ca^2+^ and PO_4_^3−^ distribution into consideration, a biological mechanism governing the local synthesis and supplementation of PO_4_^3−^, as well as its subsequent influx into matrix vesicles, must be necessary. In contrast to PO_4_^3−^ synthesis and supplementation, Ca^2+^ is abundantly present in the tissue fluid, and annexins, which are acidic phospholipid-dependent Ca^2+^-binding proteins, are assumed to serve as the Ca^2+^ channels of the matrix vesicles. Annexin A5 is the most abundant protein among annexins [46,47,48], and it appears to display Ca^2+^ channel activity in matrix vesicles. Kirsch et al. have previously reported that annexin A5 mediates Ca^2+^ influx into matrix vesicles secreted from hypertrophic chondrocytes in the growth plate, thereby initiating cartilage mineralization [49]. Thus, Ca^2+^ that is abundantly present near the matrix vesicles may enter the vesicles via annexin A5. Alternatively, to enable the influx and accumulation of PO_4_^3−^ in the vesicles, the interplay among membrane transporters and enzymes on the plasma membranes appears to be necessary.

### 2.3. Local Synthesis of PO_4_^3−^ by the Activities of TNAP and ENPP1 

One of the most important enzymes enabling mineralization is TNAP, a glycosylphosphatidylinositol anchor enzyme associated with the cell membranes of matrix vesicles and osteoblastic cells. TNAP can hydrolyze various phosphate esters, especially pyrophosphates (PPi), and is broadly recognized as a hallmark of osteoblastic cells. However, the method of PPi supplementation is important. Currently, it is believed that ENPP1 mainly supplies PPi. ENPP1 is composed of two N-terminal somatomedin B-like domains, a catalytic domain, and a nuclease-like domain. Crystalline structure analysis of ENPP1 demonstrated that the nucleotides are accommodated in a pocket formed by an insertion loop in the catalytic domain of ENPP1, implying a preference for an ATP substrate [7]. Therefore, in bone mineralization, the catalytic activity of ENPP1 may generate PPi, presumably using ATPs in the extracellular fluid. The resultant PPi is then hydrolyzed by TNAP into PO_4_^3−^. However, PPi is also known to inhibit mineralization by binding to nascent hydroxyapatite crystals, thereby preventing crystal overgrowth [50,51,52]. Hence, a balance between PPi and PO4^3−^ is important for normal bone mineralization. TNAP is not uniformly distributed on the cell membranes of osteoblasts; it was distinctly observed on the basolateral sides rather than the secretory (osteoidal) domains [37,53]. We recently demonstrated that ENPP1 was mainly localized in mature osteoblasts and osteocytes, while TNAP was seen on preosteoblasts and mature osteoblasts [54] (Figure 2). It is interesting that ENPP1, which produces PPi, a mineralization inhibitor, is localized near matrix vesicles, whereas TNAP is located on preosteoblasts and the basolateral membranes of mature osteoblasts, which are relatively distant from matrix vesicles (Figure 2). It may be important that the biological PO4^3−^ influx is balanced with the nucleation and growth of crystalline calcium phosphates inside the matrix vesicles, as well as the inhibition of ectopic mineralization of calcium phosphates deposited in areas other than the matrix vesicles, e.g., collagen fibrils without mediating matrix vesicles.

### 2.4. Transport of PPi and PO_4_^3−^ via ANK and Pit1/Pit2

ANK, encoded by the progressive ankylosis gene (*Ank*), can serve as a non-enzymatic PPi channel, allowing PPi to pass through the plasma membrane to the outside of the cell [16,17]. As shown in our recent reports, the immunoreactivity of ENPP1 was detected not only in the cell membranes but also in the cytoplasmic region of osteoblasts and osteocytes, indicating the presence of both extracellular and intracellular PPi in these cells [54]. It is therefore likely that the ANK-mediated outflow of intracellular PPi may be involved in the dynamic equilibrium between intra- and extracellular levels of PPi. After the outflow of PPi to the extracellular region, TNAP hydrolyzes PPi into PO_4_^3−^.

Extracellular PO_4_^3−^ may pass through the plasma membrane of the matrix vesicles by Pit1 and Pit2 mediation. Pit1 and Pit2 are type III sodium-inorganic phosphate (Pi) co-transporters encoded by Slc20a1 and Slc20a2 [8,9,10,11,12]. Recently, it has been reported that Pit1 and Pit2 form heterodimers, sense extracellular PO_4_^3−^ concentrations, and increase the expression of matrix Gla protein (MGP) and osteopontin via the extracellular signal-regulated kinase (ERK) pathway [55,56,57]. These reports suggest that Pit1 and Pit2 function not only as PO_4_^3−^ transporters (the influx of Pi from extracellular to intra-vesicular) but also as Pi sensors and transmit signals of the rapidly accelerated fibrosarcoma (Raf)/mitogen-activated protein kinase (MEK)/ERK pathway to synthesize bone matrix proteins [58,59]. Regarding the matrix vesicles, Pit1 and Pit2 seem to serve as transporters of PO_4_^3−^ from the extracellular fluid to intra-vesicular regions of the matrix vesicles. 

### 2.5. PHOSPHO1 for PO_4_^3−^ Production inside Matrix Vesicles

Alternative to the biological function of ENPP1/TNAP, PHOSPHO1 is an enzyme highly expressed in mineralizing osteoblasts and hypertrophic chondrocytes [60] (Figure 2). This enzyme has been implicated in bone and cartilage formation and is thought to function inside cells and matrix vesicles to generate PO_4_^3−^ using phosphocholine and phosphoethanolamine, which are components of the lipid bilayers of matrix vesicles [13,14,15]. Roberts et al. documented that PHOSPHO1 is restricted to mineralizing regions of the bone and growth plate and plays a role in the initiation of matrix vesicle-mediated mineralization [14]. Consistently, *Phospho1^−/−^* mice displayed hypo-mineralization of the bone at birth, biomechanical incompetency, scoliosis, and spontaneous greenstick fractures [61,62]. More recently, Dillon et al. demonstrated ectonucleotide phosphodiesterase/pyrophosphatase member 6 (*Enpp6*)*^−/−^* mice also revealed hypo-mineralization [63]. ENPP6 is a member of the nucleotide pyrophosphatase/phosphodiesterase family with lysophospholipase C activity, generating phosphocholine with a monoacylglycerol byproduct [64,65,66]. Therefore, ENPP6 may functionally lie upstream of PHOSPHO1 to generate intra-vesicular phosphocholine. As proposed by Stewart et al. (2018), ENPP6, as well as phospholipase A2 (PLA2), may act in sequence in the matrix vesicle membranes to produce phosphocholine, which PHOSPHO1 subsequently hydrolyzes into PO_4_^3−^ [67]. Taken together, both PLA2/ENPP6/PHOSPHO1 and TNAP/ENPP1/Pit1/Pit2 are important cascades for liberating and increasing the PO_4_^3−^ concentration in matrix vesicles. Both PLA2/ENPP6/PHOSPHO1 and TNAP/ENPP1/Pit1/Pit2 pathways may be necessary for normal mineralization in the bone.

## 3. Development of Mineralized Nodules and Collagen Mineralization

### 3.1. Growth of Mineralized Nodules

The calcium phosphate crystals that are nucleated inside the matrix vesicles grow in all directions and then penetrate the plasma membrane to exit the vesicles, eventually forming mineralized nodules, which are also referred to as calcifying globules [1,3,4]. Under TEM observation, mineralized nodules appear as globular structures composed of radially assembled hydroxyapatite crystals [44,68]. It seems likely that the growth of mineralized nodules is regulated by non-collagenous proteins in the osteoid. Among these materials, osteopontin is especially suited to regulating mineralization because it is a negatively charged and highly phosphorylated molecule that can effectively inhibit hydroxyapatite formation and growth [6,69]. Osteocalcin is another important bone matrix protein subjected to vitamin K-dependent carboxylation at its glutamate residues. Using crystal structural analysis, Hoang et al. demonstrated a negatively charged protein surface in γ-carboxylated osteocalcin, which could bind to Ca^2+^ in a hydroxyapatite crystal lattice [70]. In our experiments, the administration of warfarin, an inhibitor for the γ-carboxylation of glutamine residues, resulted in the failure of mineralized nodule formation and the dispersal of numerous needle-shaped crystal mineral fragments throughout the osteoid [71] (Figure 3). Recently, γ-carboxylase-deficient mice revealed the same abnormality with disassembled, scattered crystal minerals in the bone, which is consistent with our observation of the warfarin-treated rats [72]. When used clinically, warfarin administration to women in the first trimester of pregnancy is associated with embryopathy characterized by stippled epiphyses and distal extremity hypoplasia [73]. Therefore, osteocalcin may play an important role in the formation of the globular assembly of needle-shaped mineral crystals, i.e., mineralized nodules, which can make focal contact with collagen fibrils to enable normal mineralization in bone.

### 3.2. Collagen Mineralization

Collagen mineralization begins at the point of contact with mineralized nodules. TEM observations demonstrated that mineralization spreads from the contact point of the mineralized nodules toward the periphery of the collagen fibrils [2]. This finding suggests that collagen mineralization orderly progresses from the contact points with mineralized nodules, presumably allowing the regular deposition of calcium phosphate crystals onto the collagen fibrils. At a higher magnification, the spicules of calcium phosphate crystals can be seen on the fibrillar structures identical to the superhelix (tropocollagen) of collagen fibrils, thus indicating that mineral crystals are deposited on the superhelix, which serves as a scaffold for collagen mineralization. After contact with the mineralized nodules, the collagen fibrils eventually become completely mineralized.

Proteoglycans such as decorin and biglycan, which directly bind the collagen surface through GAG chains, inhibit the growth of mineral crystals [74,75,76]. Collagen mineralization in the osteoid increases proportionally based on the distance from the osteoblasts, whereas the amount of decorin in the osteoid decreases further away from the bone surface [77]. In the osteoid close to the osteoblasts, proteoglycans combined with the surface of newly formed collagen fibrils are localized to the large space between collagen fibrils. By contrast, in the areas close to the mineralization front of the osteoid, proteoglycans are almost exclusively bound to mature collagen fibrils and are rarely found between the narrowed spaces of collagen fibrils. Moreover, the levels of decorin mRNA and protein expression are significantly decreased before and at the beginning of matrix mineralization [78]. Therefore, collagen mineralization may also be regulated by proteoglycans. Proteoglycans not only contribute to the inhibition of bone mineralization but also modulate the aggregation of collagen fibrils [23,79,80]. Biglycan-knockout mice and biglycan/decorin double-knockout mice exhibit reduced bone mineral density as well as abnormal morphology of collagen fibrils in the bone matrix [21]. Taken together, the growth of mineral crystals and the maturation of collagen fibrils through the modulation of proteoglycans substantially affect the progression of collagen mineralization.

## 4. Osteocyte Network and the Biological Function of Regulating Bone Mineralization

### 4.1. Distribution of the Osteocyte Network 

Osteoblasts secrete bone matrix proteins and can become embedded in the bone matrix, where they differentiate into osteocytes. Immediately before becoming embedded into the bone matrix, osteoblasts rearrange the actin filament assembly along the cell membranes and the cytoplasmic processes, which resemble that of embedded osteocytes [24]. This implies that the osteoblasts approaching osteocytic differentiation and the newly-differentiated osteocytes decide the geometrical structure of the cellular network of their cytoplasmic processes. In addition, the collagen fibril orientation seems to be associated with the direction of the cytoplasmic processes of the osteoblasts and newly-differentiated osteocytes [81]. Repp et al. have documented that the intimate link between the OLCS and the collagen molecules implies an interplay between osteocyte processes and the arrangement of the surrounding collagen fibers [82]. Regularly distributed osteocytes and their cytoplasmic processes appear to control the rearrangement of mineral crystals parallel to the long axis of the collagen fibrils [83]. 

Recent review articles have shown variations in the osteocyte distribution within differently organized bone matrices during bone development and morphogenesis [84,85]. There are different types of osteogenesis (i.e., static osteogenesis (SO) and dynamic osteogenesis (DO)), which decide the fate of the osteocytes and the geometrical arrangement of the canalicular network. Although SO and subsequent DO occur in the process of intramembranous ossification, only DO occurs directly close to the remnants of the calcifying cartilage during endochondral ossification [84,85]. We thus postulated that the slowly-embedded osteocytes in the bone matrix, especially in the cortical bone that induces SO and subsequent DO, show the geometrically regulated distribution of cell bodies and cytoplasmic processes. 

Osteocytes are housed in osteocytic lacunae in the bone, while their thin cytoplasmic processes pass through narrow channels, referred to as osteocytic canaliculi, interconnected through gap junctions [86,87]. Thereby, osteocytes build two passageways: one is a cytosolic track through the cytoplasmic processes of osteocytes and osteoblasts, and the other is a narrow space between the canalicular walls and the cytoplasmic processes. It is believed that osteocytes and their meshwork of cytoplasmic processes establish a functional group for intercellular communication, molecular transport, mechanical stress-sensing, and mineralization regulation [25,26,27,28]. However, it is likely that osteocytes featuring two passageways in the OLCS communicate with other bone cells, such as osteoblasts/bone lining cells and preosteoblasts, by mediating the processes of wiring and volume transmission, as reported by Marotti [88]. This indicates that osteoblasts and osteocytes may reciprocally modulate their functions not only through volume transmission (paracrine and autocrine stimulation) but also through wiring transmission (that is, in a neuronal-like manner). In addition, recent reports have demonstrated new insights into osteocytic function, such as the piezo1 ion channel for mechanical signaling [89], the Cas–NF-κB interplay in osteocytes upon shear stress induction [90], and EphrinB2-RhoA-limited autophagy in osteocytes during secondary mineralization [91], all of which may be breakthroughs for better understanding the physiological function of osteocytes and their network.

### 4.2. Osteocyte-Derived Molecules Involved in Peripheral Mineralization 

Osteocytes physiologically synthesize several important molecules, e.g., dentin matrix protein (DMP) 1, matrix extracellular phosphoglycoprotein (MEPE), osteopontin, and Phex, for regulating surrounding bone mineralization. DMP1 has a high Ca^2+^-binding capacity and, therefore, is postulated to play a role in bone mineralization in the vicinity of osteocytes [33]. Recently, Oya et al. demonstrated that C-terminal DMP1 is phosphorylated within osteocytes and then secreted into the peri-canalicular matrix of mineralized bone, suggesting that negatively-charged phosphorylated C-terminal DMP1 plays an important role in recruiting Ca^2+^ in the peri-canalicular matrix [34]. DMP1 belongs to the small integrin-binding ligand N-linked glycoprotein (SIBLING) family, which also includes MEPE, osteopontin, bone sialoprotein, and dentin sialo-phosphoprotein, and is encoded by a gene located on human chromosome 4q21 and mouse chromosome 5q21 [92,93]. Regarding MEPE, Rowe et al. identified the novel functional domain acidic serine-rich and aspirate-rich motif (ASARM) peptide, which is highly conserved across species. MEPE can be cleaved by cathepsin B to release the carboxy-terminal ASARM peptide [31,94], which is then phosphorylated to inhibit bone mineralization [31]. However, MEPE reversibly binds to Phex, which protects MEPE from proteolysis by cathepsin B, indicating that the MEPE-Phex complex blocks the inhibition of mineralization [32]. As with osteopontin, Addison et al. demonstrated that the ASARM peptide of osteopontin inhibited mineralization by binding to hydroxyapatite in a phosphorylation-dependent manner and that Phex blocked this mineralization inhibition [95]. Since osteocytes express abundant MEPE [96], DMP1 [33], and osteopontin, especially in *Hyp* mice fed a high-phosphate diet [97], it can be easily assumed that osteocyte-derived SIBLINGs would regulate peripheral bone mineralization by the osteocytes. This postulation is evidenced by the report that a DMP1 absence results in rickets or osteomalacia in mice [98] and by autosomal recessive hypophosphatemic rickets/osteomalacia (ARHR) in human patients [99]. Hence, osteocytes seem to be involved in the regulation of the surrounding mineralization. However, Phex/SIBLINGs are usually associated with the congenital deformities rickets and osteomalacia. Therefore, it is necessary to elucidate whether SIBLINGs play an important role in the physiological regulation of bone mineralization in a normal state (Figure 4).

## 5. Cellular Interplay between Osteoblasts and Osteocytes in Bone Mineralization

Osteoblasts secrete matrix vesicles, which provide initiation sites for mineralization during primary mineralization, while osteocytes appear to regulate bone mineralization through Phex/SIBLINGs. Taking these findings into account, the interplay between osteoblasts and osteocytes in the regulation of bone mineralization seems likely. Matrix vesicles secreted by osteoblasts grow into globular assemblies of needle-like calcium phosphate crystals, called mineralized nodules, which then induce collagen mineralization. During nucleation and subsequent growth inside the vesicles, the influx of Ca^2+^ and PO_4_^3−^ is promoted by many enzymes and membrane transporters located on the matrix vesicles and mineralized nodules (particularly, they are located on the ruptured membranes of the vesicles). 

However, the growth of large, terminal mineralized nodules that are distant from osteoblasts, as well as collagen mineralization, may be regulated by a mechanism other than enzymes associated with matrix vesicles secreted by the osteoblasts. In the osteoid, there seem to be two possible pathways that supply Ca^2+^ and PO_4_^3−^ to terminal mineralized nodules and collagen mineralization: one is from the osteoblast-covered bone surface, and the other is from osteocytic canaliculi, which are opened to the osteoid. We have observed abundant ENPP1 on the secretory membranes of osteoblasts, while TNAP activity was evident on the basolateral membranes rather than the secretory side [81]. This may imply that ENPP1 provides abundant PPi beneath the osteoblasts to inhibit excessive nucleation and Ca^2+^ and PO_4_^3−^ deposition outside the matrix vesicles. In other words, matrix vesicle-mediated mineralization is initiated by the finely-tuned nucleation of Ca^2+^ and PO_4_^3−^ only inside the vesicles. 

In our study, the bones of the *kl/kl* mice displayed broad unmineralized areas despite highly concentrated serum Ca^2+^ and PO_4_^3−^ [33]. However, under higher magnification by TEM, the osteoblasts of the *kl/kl* mice produced many matrix vesicles which did not grow into large, terminal mineralized nodules [100]. Hence, supplementation with Ca^2+^ and PO_4_^3−^ from osteocytic canaliculi may be predominant compared with that from osteoblast-covered bone surfaces, or Ca^2+^ and PO_4_^3−^ supplementation from both pathways may be necessary for normal mineralization. Whichever the reason, the interplay between osteoblasts and osteocytes appears to be necessary for the sequential, finely-tuned processes of normal bone mineralization, i.e., nucleation in the matrix vesicles, formation of mineralized nodules, and collagen mineralization.

Wingless/int1 (Wnt)/β-catenin signaling may be involved in the interplay of osteoblasts and osteocytes on mineralization. Zhou et al. have demonstrated the downregulation of the Wnt/β-catenin pathway during the terminal mineralization process. They also revealed that aberrant activation of Wnt/β-catenin signaling in osteocytes resulted in the deposition of extra-large mineralized nodules on collagen fibrils [101]. Wang et al. have proposed the paradigm shift that bone mineralization is directly linked to osteocytes but not osteoblasts [76]. The authors also documented that osteocyte defects lead to the onset of osteomalacia via a sharp increase in β-catenin mechanisms. 

## 6. Conclusions

Primary mineralization in bone is achieved by matrix vesicle-mediated mineralization; matrix vesicles contain a variety of membrane transporters and enzymes involved in the nucleation and subsequent growth of crystalline calcium phosphates inside the vesicles. For proper mineralization, the biological accumulation of Ca^2+^ and PO_4_^3−^ in the vesicles is necessary. Of particular importance is the influx of PO_4_^3−^ into matrix vesicles, which involves a complex interplay among ENPP1, ANK, TNAP, and Pit1. Crystalline calcium phosphates grow radially, penetrate the vesicle membranes, and then exit the vesicles to form mineralized nodules, which are globular assemblies of needle-shaped mineral crystals. In contrast to primary mineralization, secondary mineralization increases bone mineral density, presumably due to osteocytic functions. Osteocytes appear to regulate bone mineralization, which is mediated by Phex/SIBLINGs. Thus, bone mineralization is biologically regulated by osteoblasts and osteocytes.

## Figures and Tables

**Figure 1 ijms-23-09941-f001:**
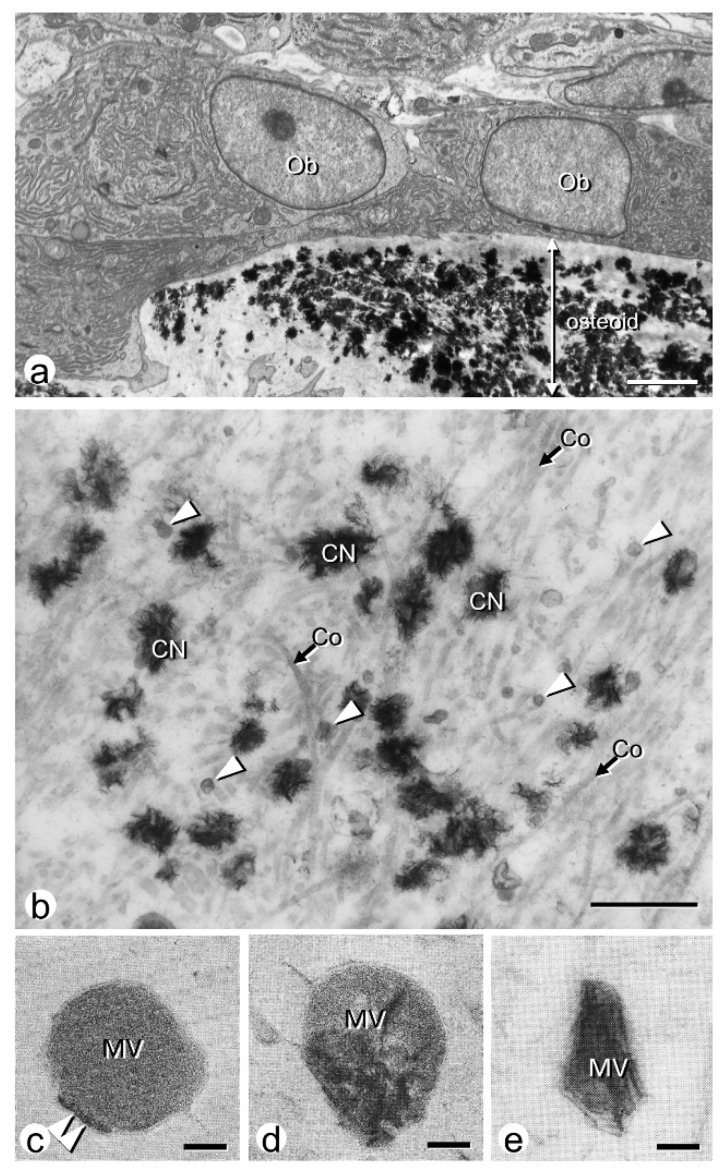
TEM observation of matrix vesicles and mineralized nodules. (**a**) Ultrastructure of osteoid underlying mature osteoblasts (Ob). (**b**) When observed at a higher magnification, there seem to be many matrix vesicles (white arrowheads), mineralized nodules (CN), and collagen fibrils (Co, arrows). (**c**) Amorphous non-crystalline phosphates (arrowheads) are observed along the inner membrane of matrix vesicles (MV) at the early stage of matrix vesicle-mediated mineralization. (**d**) The grown calcium phosphate crystals are seen inside the matrix vesicles. (**e**) The needle-like mineral crystals get out of the matrix vesicles. Panel (**a**,**b**) are cited from Ref. [19], and (**c**–**e**) are from Ref. [44]. Reprinted with permission from Ref. [19]. 2018, Springer Nature. Reprinted with permission from Ref. [44]. 1985, Japanese Association for Oral Biology. Bar, 2 mm (**a**), 1 mm (**b**), 30 nm (**c**–**e**).

**Figure 2 ijms-23-09941-f002:**
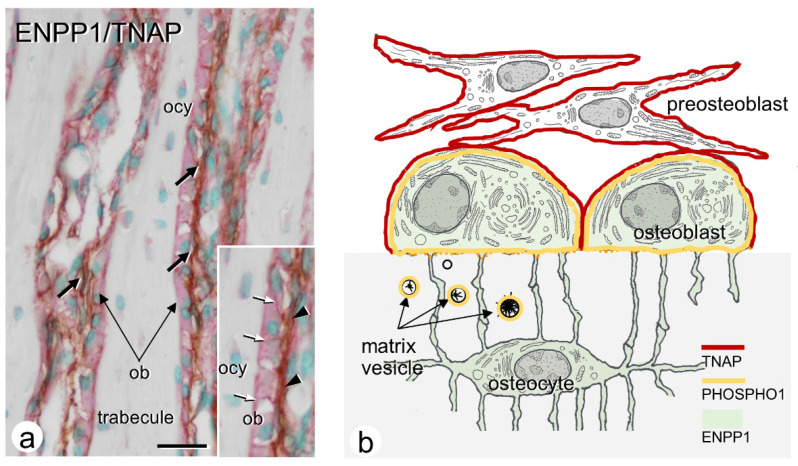
Immunolocalization of TNAP and ENPP1. (**a**) Double detection of TNAP (brown) and ENPP1 (red). Note preosteoblasts (pre-ob) and the baso-lateral sides of osteoblasts (ob) show an intense reactivity of TNAP (brown), while the cytoplasm of osteoblasts and osteocytes (ocy) reveals ENPP1 reactivity (red). (**b**) A schematic design of the distribution of TNAP (red lines), ENPP1 (green color), and PHOSPHO1 (yellow lines). Panel **a** and **b** are modified from Ref. [54]. Adapted with permission from Ref. [54]. 2021, Elsevier. Bar, 15 mm (**a**).

**Figure 3 ijms-23-09941-f003:**
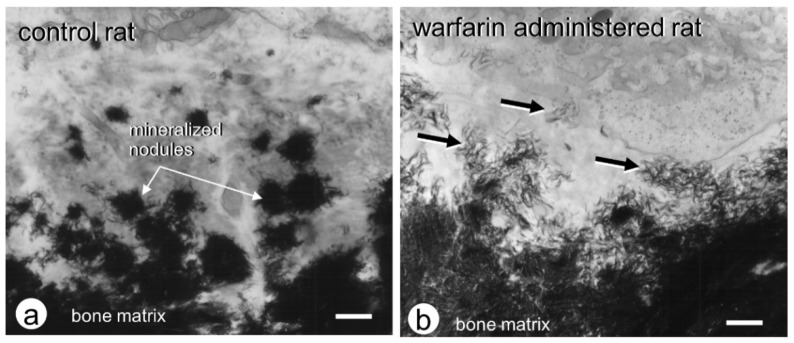
TEM observation on normal mineralized nodules and dispersed mineral crystals in osteoid. (**a**) Normal rats demonstrate globular assembly of mineral crystals in osteoid. (**b**) When administered with warfarin, an inhibitor of g-carboxylation, however, many dispersed mineral crystals (arrows) are seen in the osteoid. The images are cited from Ref. [71]. Reprinted with permission from Ref. [71]. 2009, Oxford University Press. Bar, 2 mm.

**Figure 4 ijms-23-09941-f004:**
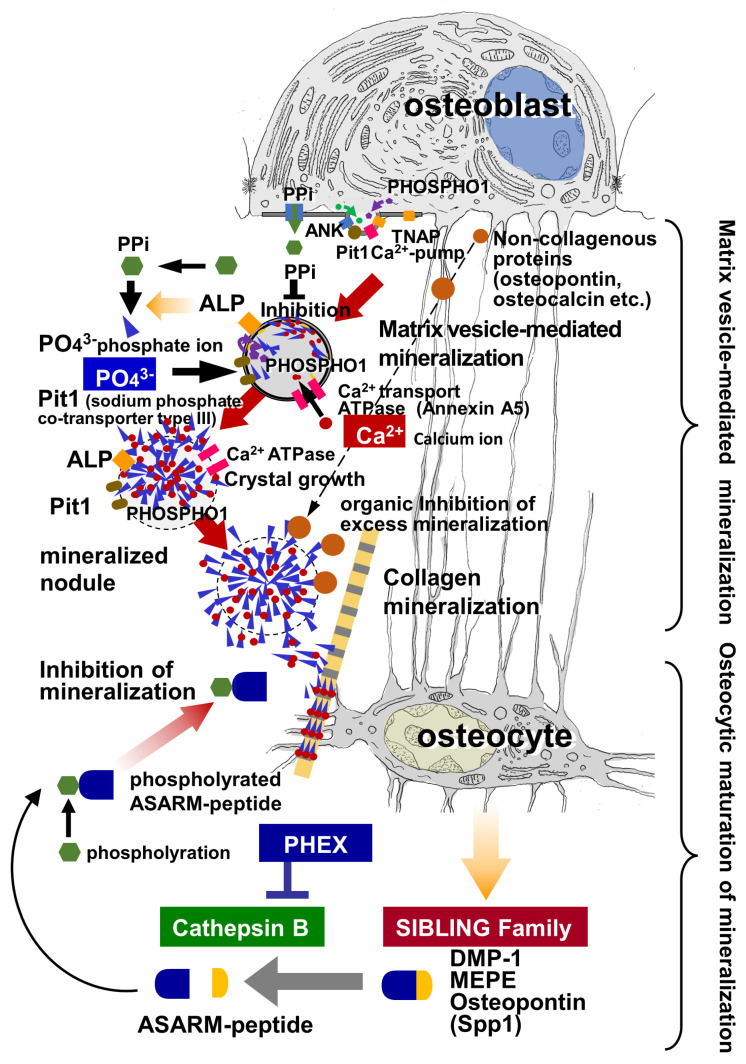
A schematic design of matrix vesicle-mediated mineralization and subsequent osteocytic maturation of mineralization.

## Data Availability

Not applicable.

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
