# Peer review of "Matrix Vesicle-Mediated Mineralization and Osteocytic Regulation of Bone Mineralization"

_ijms, 2022, doi:10.3390/ijms23179941_

Round 1

Reviewer 1 Report

In the review paper "Matrix vesicle-mediated mineralization and osteocytic regulation of bone mineralization" the authors discussed the process of bone mineralization and the role of osteoblasts and osteocytes in this process. 

The paper is very interesting, well-written and provides an overview of the current knowledge in the field. The paper is accompanied with several very informative figures which increase the readability and quality of the manuscript. 

Author Response

Our responses to the reviewer #1

Reviewer’s general comments

In the review paper "Matrix vesicle-mediated mineralization and osteocytic regulation of bone mineralization" the authors discussed the process of bone mineralization and the role of osteoblasts and osteocytes in this process.

The paper is very interesting, well-written and provides an overview of the current knowledge in the field. The paper is accompanied with several very informative figures which increase the readability and quality of the manuscript.

Our responses

We wish to express our gratitude to you for your consideration of our paper and for reviewing our manuscript. We sincerely thank you for the time and energy you spent on our manuscript and for appreciating our manuscript.

Reviewer 2 Report

July, 18th 2022

The authors investigate the role of osteocyte in the mineralization process, assuming that osteocytes enable the transport of Ca2+ and PO43− through the canaliculi of the osteocyte network, as well as remodel the surrounding bone matrix via the Phex/SIBLINGs; they conclude that bone mineralization is biologically regulated by both osteoblasts and osteocytes.

Major comments:

The work is clearly written, but some topics require reference to some morphological aspects to be adequately discussed.

Authors wrote: [lines 50-53] Hence, matrix vesicle-mediated mineralization is categorized as primary mineralization that takes place in the osteoid, which then forms a mineralized nodule, also called a calcifying nodule, allowing the collagen mineralization to eventually spread throughout the bone.

Rev: the relevant role of proteoglycans in osteoid maturation should also be mentioned.

Authors wrote: [lines 251-252] … thus indicating that mineral crystals are deposited on the superhelix, which serves as a scaffold for collagen mineralization

Rev: Authors should also considerate that, in parallel to the deposition of mineral crystal along collagen fibrils, proteoglycans (PGs) decrease in amount the further away they are from the osteoidogenic surface. Near the osteoblasts, PGs fill the wide interfibrillar space and are associated with the surface of the newly-formed collagen fibrils, whereas close to the mineralizing surface PGs are mainly associated with the almost mature collagen fibrils, and few are free within the narrow interfibrillar spaces. This datum, observed by others, support previous sentences by the authors on the role of organic materials in the osteoid mineralization [lines 218-224].

Authors wrote: [lines 67-69] Mature, well-mineralized bone develops an orderly and arranged OLCS, while immature bone features an irregular, disorganized OLCS. The author cite a paper published in 2009 [25].

Rev: the different distribution of osteocytes inside differently organized bone matrix is clearly explained in a more recent review (https://doi.org/10.3390/jfmk6010028) and depends also on the different type of bone formation, i.e. static osteogenesis versus dynamic osteogenesis (https://doi.org/10.3390/app11052025).

Authors wrote [lines 268-273]: Osteocytes are housed in osteocytic lacunae in the bone, while their thin cytoplasmic processes pass through narrow channels, referred to as osteocytic canaliculi, interconnected through gap junctions [78, 79]. Thereby, osteocytes build two passageways: one is a cytosolic track through the cytoplasmic processes of osteocytes and osteoblasts, and the other is a narrow space between the canalicular walls and the cytoplasmic processes.

Rev: What authors called “osteocytes build two passageways” corresponds to two ways by which osteocytes communicate with the other bone cells, i.e. wiring transmission and volume transmission (Marotti, 2000 - The osteocyte as a wiring transmission system. J Musculoskelet Neuronal Interact –see attached pdf file).

Authors [lines 327-358]: 4.3. Osteocytic osteolysis and lacunar-canalicular remodeling

Rev: The problem of peri-osteocyte lysis is widely discussed and recurrent. Often, if one does not have a deep understanding of the fact that the morphology of osteocytes (and therefore of the lacunae containing them) is very different in different bone matrix textures (interwoven-fiber or lamellar, and depending on the type of lamellae) one runs the risk (and it has happened before) of considering osteocyte lacunae located in woven-fibered bone "enlarged" as a result of osteocyte lysis (in a sample-control), comparing them (in a sample-treated) to osteocyte lacunae located in lamellar bone; whereas, simply, the former are wider than the latter, on the basis of  the texture of the bone in which they are located (attached is a publication that may be helpful in understanding the relief made by the reviewer: Metab Bone Dis & Rel Res, 1979). Also the authors cite, in this regard, Jandl et al. 2020.

Moreover, in a paper of Tazawa et al., 2004 [1] it is reported: <<Marotti and coworkers also pointed out that the process of osteocytic osteolysis could not be accepted as a provenphenomenon,  because  the  previous  studies  had  notmade precise reference to the types of bone tissues, such as  woven  or  lamellar  bones,  and  had  not  taken  intoaccount the direction in which the sections had been cut,which affects the sizes and shapes of osteocyte lacunae, because osteocytes and their lacunae are not spherical, but, rather, ellipsoidal>>.

[1] Kohei Tazawa, Kazuto Hoshi, Shinichiro Kawamoto, Mikako Tanaka, Sadakazu Ejiri and Hidehiro Ozawa (DOI 10.1007/s00774-004-0519-x). Osteocytic osteolysis observed in rats to which parathyroid hormonewas continuously administered. J Bone Miner Metab (2004) 22:524–529.

Authors wrote [lines 327-328]: During lactation, osteocytes reportedly erode the surrounding bone matrix by sharing a similar gene expression to osteoclasts during bone resorption.

Rev: Moreover, it is very wrong to speak of osteocyte "erosion" (the osteocyte does not possess the erosive apparatus typical of osteoclasts); what certainly can happen as a result of transient changes in calcemia (which are reflected in the composition of the fluid filling the fluid compartment of the bone) is that the osteocyte can dissolve the calcium crystals closest to its lacuno-canalicular contour, as suggested by Nango et al. 2016, cited by the authors. (Attached is a publication that may be helpful in understanding the relief made by the reviewer: Calcif Tissue Int, 1990; Bone, 1985).

Authors wrote [lines 355-358]: The authors assumed that changes in canalicular remodeling by osteocytes is a process by which the mineral is utilized from the skeleton for callus formation and bone repair following a fracture, but this process may also lessen the bone quality and elevate the fracture risk systemically [111].

Rev: New-organization in extension/interconnection of the canalicular network should not be confused with the term "remodeling" (also misleadingly used by other researchers), which indicates the more sophisticated process of the remodeling of bone structure.

Authors wrote [lines 360-364]: Osteoblasts secrete matrix vesicles, which provide initiation sites for mineralization during primary mineralization, while osteocytes appear to regulate bone mineralization by means of osteocytic osteolysis and Phex/SIBLINGs. Taking these findings into account, the interplay between osteoblasts and osteocytes in the regulation of bone mineralization seems likely. [lines 423-426]: Osteocytes appear to control mineralization and remodeling of the peripheral bone matrix, which is mediated by osteocytic osteolysis and Phex/SIBLINGs. Thus, bone mineralization is biologically regulated by osteoblasts and osteocytes.

Rev: The important thing is to differentiate the primary process of "maturation" of the osteoid and the possible osteocyte role in secondary mineralization from the osteocyte response to hypocalcemia by the dissolution (along the inner surface of the lacuno-canalicular cavities) of part of the mineral surrounding the osteocyte protoplasm (cell body and citoplasmic elongations).

Minor comments:

[lines 58-60]: Delete these parts of the paragraph: (1) “particularly in each osteon in mature cortical bone”, because there are no differences in the modes of osteoid mineralization in osteonic systems compared with other contexts (e.g., apposition bone or primary bone that does not have lamellar organization); (2) “Because of the varying formation times of each osteon, cortical bones can be seen as a mosaic work of osteons when observed using contact microradiography”, because the “mosaic work of osteons “ depends on the different "age" of the osteons, which therefore have different levels of mineralization (newer or more mature) but the modes by which mineralization proceeds are the same.

Authors, in Figure 5, compare enlarged lacunae with respect to other lacunae. 

Rev: A small magnification of the microscopic field should be shown to understand the texture of the collagen surrounding the osteocytes; otherwise, the comparison between osteocyte lacunae does not have an absolute value

Overall and final opinion of the Reviewer:

The manuscript is interesting but there are some aspects to improve before publication. The main criticality lies in the lack of morphological knowledge of the structural context in which the osteocytes live in the different locations and as a result of different ways of bone formation. From this aspect also derives the misinterpreted discussion on the enlargement of osteocyte lacunae due to erosive activity of osteocytes. Furthermore, the bibliographic entries should be integrated.

The manuscript is acceptable after major revision.

Attached are four publications to read that cannot be easily found, unlike others suggested in the context of the revision (for which the doi is reported).

Reviewer 3 Report

Combined with their results, in this review article, the authors fully discuss the matrix vesicle-mediated bone mineralization and the possible regulatory and interplay molecular mechanisms by osteocytes or osteoblasts. This is one very good review article, it will enrich the knowledge of bone biology, and is worth reading.

In this review article, firstly, the authors describe the mineralization process mediated by matrix vesicles and display some experimental pictures and evidence to support this theory. Following that, the authors cited many published experimental results to discuss the possible molecular mechanism of bone mineralization regulated and interplayed by osteocytes and osteoblasts. This is an interesting review article, the references cited are supported the authors' opinion and the manuscript writing is well and easy to read, it will benefit people to further understand the matrix vesicle theory of bone mineralization. Suggestion: If the authors could simply mention other mechanisms of bone mineralization in this manuscript will be helpful for readers to understand bone mineralization knowledge.

Author Response

Our responses to the reviewer #3

Reviewer’s general comments

Combined with their results, in this review article, the authors fully discuss the matrix vesicle-mediated bone mineralization and the possible regulatory and interplay molecular mechanisms by osteocytes or osteoblasts. This is one very good review article, it will enrich the knowledge of bone biology, and is worth reading.

In this review article, firstly, the authors describe the mineralization process mediated by matrix vesicles and display some experimental pictures and evidence to support this theory. Following that, the authors cited many published experimental results to discuss the possible molecular mechanism of bone mineralization regulated and interplayed by osteocytes and osteoblasts. This is an interesting review article, the references cited are supported the authors' opinion and the manuscript writing is well and easy to read, it will benefit people to further understand the matrix vesicle theory of bone mineralization.

Our responses

Thank you for your invaluable suggestions on our study. We agree with your suggestions. Kindly find below our point-by-point responses to each of your comments. We believe that your helpful suggestions have helped us significantly improve our paper.

Reviewer’s specific comments

Suggestion: If the authors could simply mention other mechanisms of bone mineralization in this manuscript will be helpful for readers to understand bone mineralization knowledge.

Our responses

Thank you for your suggestion.  We agree with the reviewer’s suggestion; we have added the following description of the regulation of bone mineralization by proteoglycans in the revised manuscript. Please review the pertinent corrections below.

  1. Introduction

[lines 39-42]

Osteoblasts secrete a large amount of collagen fibrils, non-collagenous proteins, and proteoglycans as well as matrix vesicles into the incompletely mineralized superficial layer of the bone matrix known as osteoid.

[lines 53-55]

Moreover, non-collagenous proteins and proteoglycans in osteoid regulate mineralization by modulating the aggregation of collagen fibrils and mineral deposition during primary mineralization [20-23].

3.2. Collagen mineralization

[lines 247-263]

Proteoglycans such as decorin and biglycan, which directly bind the collagen surface through GAG chains, inhibit the growth of mineral crystals [78-80]. Collagen mineralization in the osteoid increases proportionally based on the distance from the osteoblasts, whereas the amount of decorin in the osteoid decreases further away from the bone surface [81]. In the osteoid close to the osteoblasts, proteoglycans combined with the surface of newly formed collagen fibrils are localized to the large space between collagen fibrils. By contrast, in the areas close to the mineralization front of osteoid, proteoglycans are almost exclusively bound to mature collagen fibrils and are rarely found between the narrowed spaces of collagen fibrils. Moreover, the levels of decorin mRNA and protein expression are significantly decreased before and at the beginning of matrix mineralization [82]. Therefore, collagen mineralization may also be regulated by proteoglycans. Proteoglycans not only contribute to the inhibition of bone mineralization but also modulate the aggregation of collagen fibrils [23, 83, 84]. Biglycan-knockout mice and biglycan/decorin double-knockout mice exhibit reduced bone mineral density as well as abnormal morphology of collagen fibrils in the bone matrix [21]. Taken together, the growth of mineral crystals and the maturation of collagen fibrils through the modulation of proteoglycans substantially affects the progression of collagen mineralization.

References

[20]  Xu, S.; Bianco, P.; Fisher, LW.; Longenecker, G.; Smith, E.; Goldstein, S.; Bonadio, J.; Boskey, A.; Heegard, AM.; Sommer, B.; Satomura, K.; Dominguez, P.; Zhao, C.; Kulkarni, AB.; Gehron, Robey, P.; Young, MF. Targeted disruption of the biglycan gene leads to an osteoporosis-like phenotype in mice. Nat. Genet. 1998, 20, 78–82.

[21]  Corsi, A.; Xu, T.; Chen, XD.; Boyde, A.; Liang, J.; Mankani, M.; Sommer, B.; Iozzo, RV.; Eichstetter, I.; Robey, PG.; Bianco, P.; Young, MF. Phenotypic effects of biglycan deficiency are linked to collagen fibril abnormalities, are synergized by decorin deficiency, and mimic Ehlers-Danlos-like changes in bone and other connective tissues. J. Bone. Miner. Res. 2002, 17, 1180-1189. 

[22]  Kemp, LP.; Morris, JA.; Medina-Gomez, C.; Forgetta, V.; Warrington, NM.; Youlten, SE.;  Zheng, J.; Gregson, CL.; Grundberg, E.; Trajanoska, K.; Logan, JG.; Pollard, AS.; Sparkes, PC.;  Ghirardello, EJ., Allen, R.; Leitch, VD.; Butterfield, NC.; Komla-Ebri, D.; Adoum, AT.;  Evans, DM. Identification of 153 new loci associated with heel bone mineral density and functional involvement of GPC6 in osteoporosis. Nat. Genetics. 2017, 49, 1468-1475.

[23]  Hao, JX.; Shen, MJ.; Wang, CY.; Wei, JH.; Wan, QQ.; Zhu, YF.; Ye, T.; Luo, ML.; Qin, WP.; Li, YT.; Jiao, K.; Zhao, B.; Niu, LN. Regulation of biomineralization by proteoglycans: From mechanisms to application. Carbohydr. Polym. 2022, 294, 119773.

[78]  Boskey, AL.; Spevak, L.; Doty, SB.; Rosenberg, L. Effects of bone CS-proteoglycans, DS-decorin, and DS-biglycan on hydroxyapatite formation in a gelatin gel. Calcif. Tiss. Int. 1997, 61, 298-305.

[79]  Tavafoghi, M.; Cerruti, M. The role of amino acids in hydroxyapatite mineralization. J. R. Soc. Interface. 2016, 13, 20160462.

[80]  Wang, K.; Ren, Y.; Lin, S.; Jing, Y.; Ma, C.; Wang, J.; Yuan, XB.; Han, X.; Zhao, H.; Wang, Z.; Zheng, M.; Xiao, Y.; Chen, L.; Olsen, BR.; Feng, JQ. Osteocytes but not osteoblasts directly build mineralized bone structures. Int. J. Biol. Sci. 2021, 17, 2430-2448.

[81]  Hoshi, K.; Kemmotsu, S.; Takeuchi, Y.; Amizuka, N.; Ozawa, H. The primary calcification in bones follows removal of decorin and fusion of collagen fibrils. J. Bone Miner. Res. 1999, 14, 273-280.

[82]  Mochida, Y.; Parisuthiman, D.; Pornprasertsuk-Damrongsri, S.; Atsawasuwan, P.;  Sricholpech, M.; Boskey, AL.; Yamauchi, M. Decorin modulates collagen matrix assembly and mineralization. Matrix Biol. 2009, 28, 44-52.

[83]  Raspanti, M.; Viola, M.; Forlino, A.; Tenni, R.; Gruppi, C.; Tira, ME. Glycosaminoglycans show a specific periodic interaction with type I collagen fibrils. J. Struct. Biol. 2008, 164, 134-139.

[84]  Orgel, JP.; Eid, A.; Antipova, O.; Bella, J.; Scott, JE. Decorin Core protein (Decoron) shape complements collagen fibril surface structure and mediates its binding. PLoS. ONE. 2009, 4, e7028.

Once again, we would like to sincerely thank you for your kind suggestions, which we believe have greatly improved the quality of our m

Round 2

Reviewer 2 Report

On the base of modifications made by the authors, the manuscript improved significantly compared to the original version. Appreciated the integrations about proteoglycans as well as SO and DO osteogenesis; also appreciated deletion about osteocytic osteolysis. Good also the changes introduced in the references.

Minor suggestion:

At line 73, enter the explanation of the acronym OLCS in the sentence “Osteocytes with well-organized OLCS in mature, but not … “, since theprevious sentence in line 69 <<osteocytic lacunar canalicular system (OLCS)>> was deleted.